# The effects of probabilistic context inference on motor adaptation

**Dario Cuevas Rivera** [1,2] *, **Stefan Kiebel** [1,2]

**1** Chair of Cognitive Computational Neuroscience, Faculty of Psychology, Technische Universität Dresden, Dresden, Germany, **2** Centre for Tactile Internet with Human-in-the-Loop (CeTI), Dresden, Germany

* dario.cuevas_rivera@tu-dresden.de

**Data Availability Statement:** All code required to reproduce results can be found in the following repository: 10.5281/zenodo.7979250.

**Funding:** Funded by the German Research Foundation (DFG, Deutsche Forschungsgemeinschaft) as part of Germany's Excellence Strategy – EXC 2050/1 – Project ID

## Abstract

Humans have been shown to adapt their movements when a sudden or gradual change to the dynamics of the environment are introduced, a phenomenon called motor adaptation. If the change is reverted, the adaptation is also quickly reverted. Humans are also able to adapt to multiple changes in dynamics presented separately, and to be able to switch between adapted movements on the fly. Such switching relies on contextual information which is often noisy or misleading, affecting the switch between known adaptations. Recently, computational models for motor adaptation and context inference have been introduced, which contain components for context inference and Bayesian motor adaptation. These models were used to show the effects of context inference on learning rates across different experiments. We expanded on these works by using a simplified version of the recently-introduced COIN model to show that the effects of context inference on motor adaptation and control go even further than previously shown. Here, we used this model to simulate classical motor adaptation experiments from previous works and showed that context inference, and how it is affected by the presence and reliability of feedback, effect a host of behavioral phenomena that had so far required multiple hypothesized mechanisms, lacking a unified explanation. Concretely, we show that the reliability of direct contextual information, as well as noisy sensory feedback, typical of many experiments, effect measurable changes in switching-task behavior, as well as in action selection, that stem directly from probabilistic context inference.

## Introduction

It has been shown that humans can adapt motor commands to counteract changes in the dynamics of the environment and their own bodies, such as performing reaching movements while holding a handle that exerts nonlinear forces [1]. This is known as motor adaptation. Moreover, human participants have been shown to adapt to different, even opposing, changes during the course of a single experiment [2, 3]. Additionally, humans have been shown to dynamically switch between different learned adaptations [1, 4–6].

By introducing blocks of trials in which body dynamics are altered (e.g. a mechanical arm exerts a force on the participant's hand), experimenters are able to observe motor adaptation through the lens of motor error. Across many different motor adaptation experiments [1, 2, 7],

390696704 – Cluster of Excellence "Centre for Tactile Internet with Humanin-the-Loop" (CeTI) of Technische Universit¨at Dresden. The funders had no role in study design, data collection and analysis, decision to publish, or preparation of the manuscript.

well-established phenomena have been observed: (i) the ability to recall previously-learned skills upon re-exposure to a previous altered dynamic, called savings; (ii) the ability to return to unmodified dynamics, termed de-adaptation; (iii) the interference in motor learning between opposing manipulations in dynamics, called anterograde interference; (iv) spontaneous display of behavior consistent with a previously-learned adaptation, during trials where errors are forced to be zero, called spontaneous recovery.

To explain these phenomena, a number of computational models have been introduced, in which motor commands are adapted based on observed motor errors. The most well-studied models are linear learners [8–10], while Bayesian accounts have also been shown to provide an alternative explanation for savings and quick de-adaptation in the form of switching between forward models [11, 12].

While these general models of adaptation explain the most common phenomena observed in experiments, other known phenomena remain outside of their scope. For example, it is known that adaptation rate is reduced in situations where the environment is unstable and unpredictable [13], or situations in which errors are small [14] or adaptations slowly introduced [15]. Action selection has also been found to depend on the history of adaptations learned [1, 16].

Recently, a new computational model for context-dependent motor learning based on Bayesian inference was introduced in [17], called Contextual Inference (COIN). Heald et al. [17] formalized context inference as a process that operates independently from motor learning, but both informs and is informed by it. With this model, the authors showed that context inference causes the observed changes in the rate of motor learning in previous experiments [18], among other phenomena.

In this work we show that the process of context inference underlies more behavioral phenomena than previously shown. We believe that these phenomena stem directly from the process of context inference under uncertainty in a Bayesian setting, as done in [17], and not from the specific mathematical form of the generative model used. To show how this Bayesian approach accounts for these phenomena, we used a minimal model for motor adaptation that includes context inference, which we derived by simplifying the COIN model (henceforth called sCOIN). We focused on the effects of uncertain contextual information on switching behavior, especially during error-clamp trials, in which errors are forced to zero by experimenters. More specifically, we focused on the effects of perceptual noise, as well as feedback modalities, in context inference, which in turn affects behavior in ways that can be directly measured. We show that through context inference, switching behavior can display three main effects that have been previously attributed to hypothesized ad-hoc mechanisms: (1) The size of an adaptation dictates how quick and reliable switching between tasks is [12, 19], which we explain in terms of the effects of perceptual noise on context inference. (2) Previously-learned adaptations can interfere with switching behavior [1], which we explain in terms of uncertain context inference. (3) Training history (i.e. which adaptations have been learned and for how long) affects switching during error-clamp trials [16], which we also attribute to uncertain context inference. To do this, we used the sCOIN model to simulate the experimental setups and the decision-making agents (i.e. participants) during those experiments.

Importantly, the goal of this work is not to introduce a new model for contextual motor learning, but to use the existing ideas of the COIN model to show that context inference can explain more experimental phenomena than those explored in [17].

With these combined simulations and the qualitative comparison to the experimental phenomena outlined above, we provide further evidence that context inference is a single, coherent and mechanistic account that underlies experimentally well-established motor adaption and history effects under changing contexts.

## Results

Using the sCOIN model, we simulated representative experiments from a number of experimental studies on motor adaptation to illustrate how this model explains different experimental findings using the dynamics of context inference. We will present these simulations alongside the experimental results from the representative studies and discuss in detail how context inference explains the experimental phenomena.

Before presenting these results, we briefly describe the COIN model and the simplifications that led to the sCOIN version used in simulations. We leave a more thorough explanation of the models for the Methods section. Additionally, we present simulations using the sCOIN model that show the effects of contextual cues and perceptual noise on context inference, which pave the way for the simulated experiments that we show later on.

### Experimental designs

In the following sections, we discuss the experiments done in [1, 12, 16, 19]. In all experiments, participants were asked to do reaching or shooting movements from a starting position to a known target, and the experimental manipulations can be classified into two categories: (1) curl forces with mechanical arms, and (2) visuomotor rotations. In this section, we briefly describe these two manipulations and leave the details of the experimental design to the sections below where each study is discussed.

In the experiments with mechanical arms [1, 16], participants performed the movements while holding the handle of a mechanical arm; the arm exerted a force on the handle on each trial which depended on the speed of the participant's hand, creating a force that was perpendicular to the direction of movement, called a curl force (see Fig 1A). In these experiments, the participants' view of their hand and the mechanical arm was blocked, and they had to rely on a representation of their hand on a screen in front of them, which included the starting position and the target. The experiments in [1, 16] provided an accurate position of the participant's hand, with no noise added to it, and the cursor representing the hand is visible throughout the experiment; note that in [16] there was noise added to the cursor on the screen during their third experiment, but only their first experiment is simulated here.

When the mechanical arm was set to exert no force, movements made by human participants were close to straight lines between the starting position and the target. When the curl force was applied, movements deviated from this straight-line path, creating curved movements that nevertheless reached the target, as participants adjusted their trajectory before the end of the movement. As participants repeated the movements over many trials, they learned to adapt to these curl forces and went back to performing near straight-line movements. During error-clamp trials, the mechanical arm engaged a very stiff spring that caused participants' movements to be a straight line, regardless of the participants' applied forces (see Fig 1A). This allowed experimenters to present a zero-error feedback to the participant, while at the same time measuring the forces participants applied onto the spring.

Because participants could see the cursor, they adapted their movements in-flight, always reaching the target. Errors were measured as deviations from the straight line connecting starting and target positions.

In the experiments with visuomotor rotations Kim et al. [19] and Oh and Schweighofer [12], participants moved a cursor on a screen by using a joystick [19], or on the surface of a mirror using the pen of a digitizer pad [12]. Movements were done from a starting position in the middle of the screen to targets appearing on a circumference of 10cm in radius. The experimental manipulation took the form of a rotation between the hand movements and the cursor of the screen (see Fig 1B), centered on the starting position. A clockwise rotation of 20 degrees

A

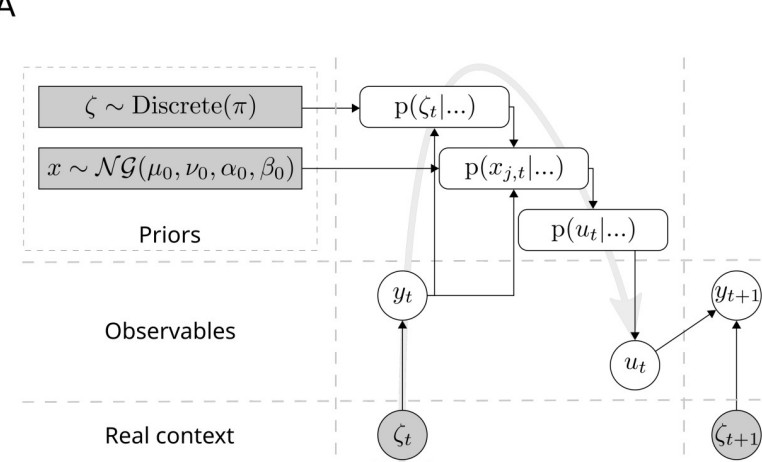

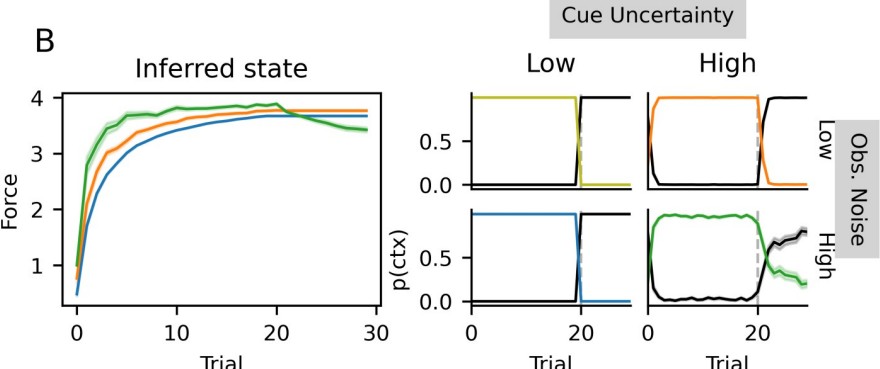

**Fig 1. Mechanical arm and visuomotor rotation experiments.** Grey and blue circles represent the starting point and the target, respectively. Targets are placed on a circumference in both experimental paradigms. (A) Mechanical arm experiment. During adaptation trials, the force exerted by the arm on the handle pushes the hand away from the straight line (i.e. the force is perpendicular to the direction of movement). In error-clamp trials, the mechanical arm creates resistance against movements perpendicular to the straight line to the target. (B) Visuomotor rotation experiment. The correspondence between hand movements and the cursor on the screen (hand icon) is rotated (visuomotor rotation). The error (brown line) is measured as the arc between the target and the place at which the trajectory (dotted line) crossed the circumference.

means that a hand movement forward is translated into a forward movement angled 20 degrees towards the right on the screen or mirror. In [19], the authors used the convention that clockwise rotations are negative, while in [12] the opposite was used.

In [12], the authors added Gaussian noise to the visuomotor rotation at each trial. With this, each trial had a different visuomotor rotation, centered at 10 or 20 degrees, and with a standard deviation of 0.5 or 4 degrees, depending on the condition. In this work, we focused on the comparison between conditions 1a and 3, with means of 20 and 10 degrees, respectively, and a standard deviation of 0.5 degrees.

In contrast to experiments with a mechanical arm, movements in visuomotor rotation experiments are always close to a straight line. Errors are measured in degrees, as the deviation between the target and the point in the 10cm circumference where the cursor crossed it (see Fig 1B).

## Modeling context-dependent adaptation

We focused on three main components of the COIN model: (1) context inference, (2) motor adaptation and (3) action selection. The processes defined by these components occur in this order, and each component informs the ones that follow.

Central to the model is the concept of context, defined in terms of the task to be performed, the variables of the environment that are relevant to perform the task, the forward models used by the decision-making agent to perform the task, and the update mechanisms necessary to adapt these forward models to the changing environment. Together, these elements allow the agent to make predictions on future observations when this context is active, and these predictions are used to infer the context. For example, when lifting an object of unknown weight, an agent might have learned one context for heavy objects and one for light objects. When observing an object to be lifted, the agent can use its size and texture to estimate the weight of the object, which in turn allows the agent to infer the appropriate context and, with it, decide how to lift the object.

The COIN model contains, additionally to these three main components, components for learning new contexts (i.e. inferring the existence of a new context that had not been previously encountered by the agent), as well as the ability to infer subject-specific parameters such as a participant's assumed transition probabilities between contexts, which can differ from the real, hidden transition probabilities. Because we seeked to focus on switching behavior between previously-learned contexts, as well as in the perceptual aspect of context inference, we chose to fix the participant-inferred transition probabilities between contexts, as well as the total number of contexts; in our simulations, we assume that participants have already inferred the real values of these quantities. Because we mainly focus on switching behavior, as well as error-clamp trials (both of which involve already-learned adaptations), these simplifications to the model have minimal effects on our results. See Methods for more details on the sCOIN model, as well as the differences between COIN and sCOIN.

By fixing the aforementioned values, the sCOIN model has a simpler generative model which allows the agent to perform exact Bayesian inference for motor adaptation. The inference process can be seen in Fig 2A, including the priors for both context inference and motor adaptation. For more details on these choices, see the Methods section.

## Contextual cues and feedback

The behavioral phenomena which are the focus of this work can be explained as arising from the effects of contextual cues and sensory feedback provided to participants during the experiment. To illustrate these effects in a simple example, we first simulated a generic motor adaptation experiment similar to those performed in [1]: participants performed reaching movements from a central position to fixed targets on a circumference, then back to the central position. All targets were positioned on the horizontal plane, at shoulder height. Movements were performed holding a robotic arm which exerted velocity-dependent rotary forces, which, when ignored, would move the participant's hand away from the straight-line path to the target. See the following section for more details on the experiments in [1]. The results of these simulations can be seen in Fig 2B.

The key to an intuitive understanding of the results presented below is to observe what happens when the presence and reliability of contextual cues is varied, as well as the perceptual noise on the position of the hand. To do this in our simulations, we added noise to the categorical variable that encodes contextual cues, and to the observations (see Methods for details), at two different levels (low and high noise). We simulated 50 participants, with 30 trials per participant. In Fig 2B, a 2x2 grid of results is shown: each simulation in this grid is a combination

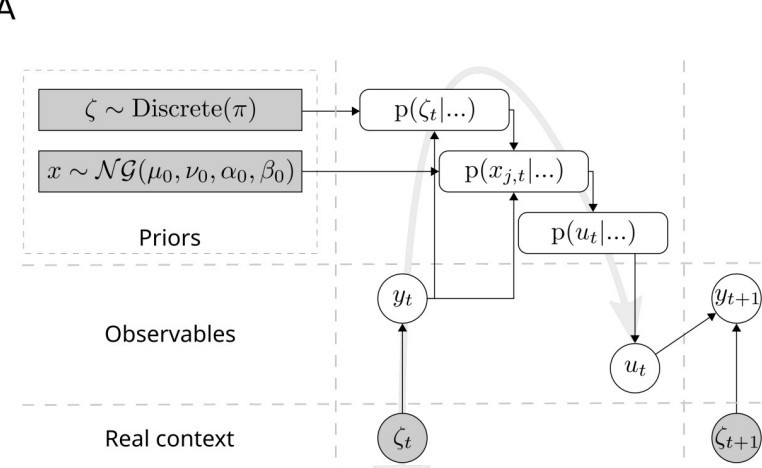

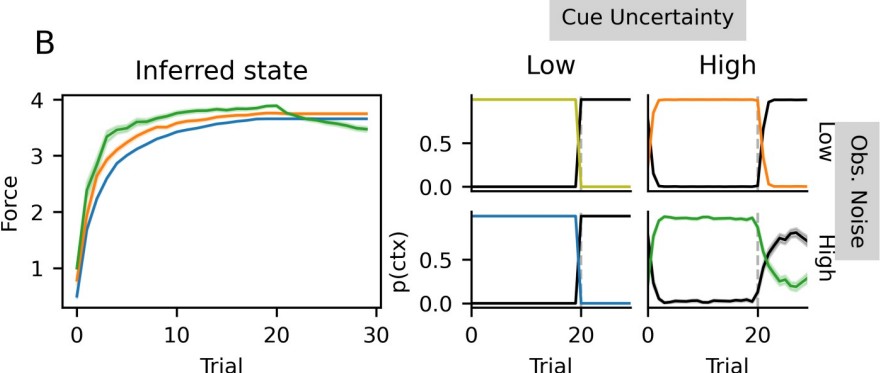

**Fig 2. Schematic representation of the sCOIN model and illustrative simulations.** (A) Inference done by the model at every trial. Clear circles represent the observables, i.e. the information available to the model (and the brain), and used for adaptation: motor commands $u_t$ and direct observations $y_t$ (e.g. cursor position). The true context $\zeta_t$ is not directly observable, but influences $y_t$. The dark rectangles represent the prior distributions for the inferred adaptation level $x_t$ (Normal-Gamma distribution) and the current context $\zeta_t$ (discrete distribution with known $\pi$; see Methods). At every trial, the context is inferred, then motor adaptation is carried out and finally a motor command is issued; the flow of this process is indicated by the gray arrow in the background, while black arrows show the direction of information flow. (B) Simulations obtained with the model in (A), using a simulated experimental setup similar to that in [1], in which the context changes at trial 20 (vertical, dashed lines). A total of 2x2 experiments were simulated, with low and high levels of both cue uncertainty and observation noise. In the left panel, the states inferred by the model for each of the 2x2 simulated experiments, where each color represents one experiment (mean ± SEM, 50 simulations per condition). On the right, each plot represents context inference $p(ctx)$ for one specific level of cue uncertainty and observation noise, with the same colors as the panels on the left. The y-axis represents the posterior probability of each context $p(\zeta_t = j)$; the black line represents the baseline context (i.e. no adaptation), the colored line (with the same colors as the panel on the left) represents the only adaptation to be learned during the simulated experiment.

of low or high contextual cue uncertainty (where high uncertainty is equivalent to presenting no cues), and low or high perceptual noise (representing how well participants can detect deviations from the straight-line movements).

Fig 2B shows that, in the presence of reliable contextual cues, context inference is accurate, certain and fast to switch; this can be seen in the context inference panel for low cue uncertainty, where the probability of the baseline context (black line) goes from close to zero to one at trial twenty. However, as contextual cues become less reliable, switching between known contexts becomes slower, as seen in the slow change in the posterior probabilities over contexts

at and after trial 20 in the last column. Furthermore, as perceptual noise increases, switching becomes not only slower, but also more uncertain, with individual agents incorrectly missing the switch entirely. While cue uncertainty and observation noise have an effect on the motor adaptation process, as seen in the difference between the different colored lines on the left-most panel in Fig 2B, in all simulations the hidden state (i.e. the force exerted by the mechanical arm, $x_{j,t}$) is quickly inferred, taking around 10 trials to reach a plateau close to the real force. After the switch back to the baseline context at trial 20, the hidden state of other context (yellow, orange, or blue, depending on the condition) is no longer updated, but selected actions become consistent with the baseline context (i.e. zero compensatory force). An exception happens in the condition with high cue uncertainty and high observation noise, in which the green context has a relatively high probability after the switch, which causes a small decay of the inferred state.

As we show below, these effects are at the heart of the behavioral phenomena observed in the experiments in [1, 12, 16, 19], and our simulated data qualitatively matches their results, as well as others that we discuss in the Discussion section.

## Experimental results

In this section, we present experimentally-observed phenomena in three sections, and show that the dynamics of context inference provide a unifying explanation for all of them. In the first section, we discuss switches between contexts, and how slow context inference (i.e. uncertainty about the real context remains relatively high over many trials) affects these switches. In the second section, we focus on interference between learned adaptations. Finally, in the third section we discuss context inference during error-clamp trials, and its effect on behavior. For each of the three sections, we selected one or two studies which are representative of the phenomenon being discussed.

For clarity, we first introduce necessary terminology that is typically used in experimental studies. As an example, we will use a typical motor adaptation task in which participants have to make reaching movements while holding the handle of a mechanical arm that exerts a curl force on the participant's hand. Depending on the trial, the mechanical arm might exert a curl force in a clockwise or counter-clockwise direction, or no force at all. Let us define the baseline context $O$ as that in which the mechanical arm exerts no force. Contexts $A$ and $B$ can be defined as those with clockwise and counter-clockwise forces, respectively. Abusing notation, a usual statement is that $B = (-A)$, as the forces have the same magnitude but point in opposite directions. Similarly, one can define context $A/2$, with the same direction of adaptation as $A$, but half the magnitude. Finally, many experiments include a block of error-clamp trials at the end of the experiment, in which the mechanical arm forces the participant's hand into a straight-line trajectory towards the target by counteracting any lateral forces, eliminating any error for the trial; we represent these with the letter $E$.

With this terminology, a typical experiment [4] would have a block structure of $O - A - B - E$, or $O - A - (-A) - E$, which means that the participant goes through a block of trials with no external force applied ($O$), a number of trials with a clockwise curl force ($A$), a block with counter-clockwise forces ($B$), and finally a block with error-clamp trials ($E$). When participants are exposed to the same context multiple times [12], an experiment can be described as $O_1 - A_1 - O_2 - \ldots$, where $A_n$ is context $A$, presented to the participant for the n-th time.

**Cue- and sensory feedback uncertainty affects switching behavior.** The term 'savings' refers to the ability to remember a previously-learned adaptation, observed as an instant recall of the previously-learned adaptation [12, 19], or as accelerated re-learning [20]; in the case of instant recall, the phenomenon is often called 'switching'. Savings are almost universally

observed in humans [3, 8, 21–23]. In an $O - A - O - A$ experiment, for example, savings would express themselves in the second A block in the form of a much higher adaptation rate than that observed during the first $A$ block. The related concept of quick de-adaptation occurs in $A - O$ or $A - (-A) - O$ transitions, where participants switch back to baseline without having to re-learn it [4].

In this section, we discuss savings in terms of switching between contexts. We show that through context inference and how it is affected by contextual cues and observation noise, savings are not immediate, but a relatively fast process that reflects context inference. We show that the manifestations of savings on behavior are mediated by context inference, which could mask the presence of savings in cases where observations do not unequivocally identify a context. Additionally, we show how the process of context inference can explain both immediate switching between known contexts and slower switches, which take the form of a high learning rate (higher than the first time a context was encountered).

To show this, we examined multiple experimental studies in which savings are observed. We categorized these studies based on the amount of contextual information made available to participants: In some experiments [5, 19], the context is clearly revealed to the participant using sensory cues. We call these cued-context experiments. In other experiments, partial information is available to participants [1, 23] in the form of large prediction errors, partial contextual information or reward prediction errors; we refer to these as partially-cued experiments.

We selected three representative experiments from two studies [12, 19] which differ in the amount of contextual information available to participants. Kim et al. [19] performed a cued-context visuomotor rotation experiment with three contexts with different rotation: no rotation, 40 degrees and -40 degrees. Participants performed shooting movements in blocks of trials with the same rotation. Importantly, a colored light identified the current context, making this a cued-context experiment. Consistent with the context-inference account, the authors found that switching was immediate and accurate.

In Fig 3A we show the results of simulations with the model using the parameters of the task, as well as the experimental results from [19]. The correspondence between simulations and experimental results can be seen in the switches between contexts in the third and fourth columns, where the solid gray line (representing the true context) switches between 40, 0 and -40, and the thick black line, representing the agent's adaptation, quickly follows these switches. For a more direct comparison between experimental results and simulations, see Fig 4.

In the second panel of Fig 3A, we show the inferred state (i.e. the inferred angle of rotation). Critically, it can be seen for the first three context exposures (until about trial 200) that participants had not yet completely adapted to the rotation, as also evidenced by their responses not being on par with the true rotation angle. This undershooting of responses (i.e. not doing the 40 degree rotation) happens despite participants being able to immediately and with high certainty identify the true context, as shown in the first-column panel. These results are similar to those shown in [24], where sensory cues of varying reliability effected immediate or slow contextual switches.

To expand on these results, we now turn to feedback and its effects on switching behavior. In [12], the authors performed two partially-cued experiments with a visuomotor rotation of 20 and 10 degrees, respectively. The results of their experiments can be seen in Fig 3B and 3C, fourth column, alongside simulations with the sCOIN model in the first three columns. Participants in the first experiment (Fig 3B) first learned the rotation in $A$. In subsequent context transitions, participants and simulations showed immediate switching (with a one-trial lag) between $A$ and $O$ (both ways), as can be seen in the posteriors over context (Fig 3B, first

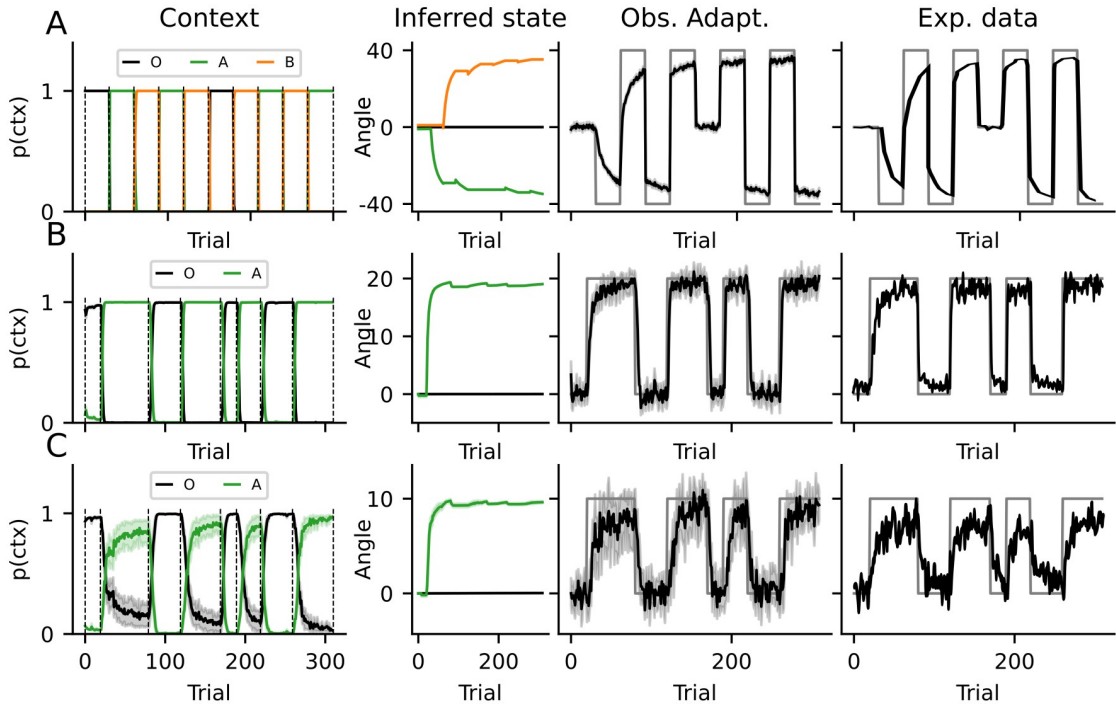

**Fig 3. Switching between learned adaptations gated by context inference.** Data from our simulations (first three columns) compared to data adapted from figure 2A in [19] and figure 4A from [12] (last column). Experimental and simulated data were, in all three experiments, averaged across all participants (20, 11 and 11 participants, respectively). In the first column, the simulated context inference is represented by the posterior probabilities over all available contexts. The colors black, green and orange represent the *O*, *A* and *B* contexts, respectively in the first two columns. Vertical, dashed lines represent switches in the real context. The inferred state (angle of the visuomotor rotation) is shown in the second column, with the same colors as in the first column. In the third and fourth columns, the black line represents the adaptation (i.e. response) displayed by the agent as a function of trial number. The thin gray lines represent the optimal adaptation, i.e. the size of the true visuomotor rotation during the task. (A) Experiment in [19]. Both experimental and simulated results are shown only up to trial 300, of the original 600. (B) An experiment in [12]. (C) Same experiment as (B) but with a 10 degree adaptation.

column, and in their responses (black line in the last two columns of Fig 3B) closely following the switches in the true rotation. For a closer comparison of simulations and experimental results, see Fig 4.

In the second experiment (Fig 3C), context switching happens more slowly, with adaptation lagging behind the switches in the real context, and slowly catching up over trials. As can be seen in the first column in Fig 3B and 3C, the same model parametrization produces fast, accurate switches when the adaptation is large (B), and slow, noisy switches when it is low (C). This difference is explained in our simulations in terms of the size of the true rotation: the posterior over contexts $p(\zeta_i|s_t, \ldots)$ (seen in Fig 3C, first column) becomes more uncertain because the errors caused by the unlearned rotation start being of the same magnitude as the observation and motor noise. In this case, uncertainty in the posteriors means that $p(\zeta_i|s_t, \ldots) \ll 1$, where $\zeta_i$ is the true context, while $p(\zeta_j|s_t, \ldots) \gg 0, \forall j \neq i$.

In contrast, in [12], the authors explained the results of their second experiment by positing that when adaptations were small, participants did not identify this as a new context and opted instead for a modification of their baseline model (i.e. how they move normally). Under this single-context explanation, however, savings do not exist, and adaptations need to be learned anew every time a context changes. On the other hand, savings are present under the sCOIN model, their manifestations being diminished by the slower context inference. Oh and

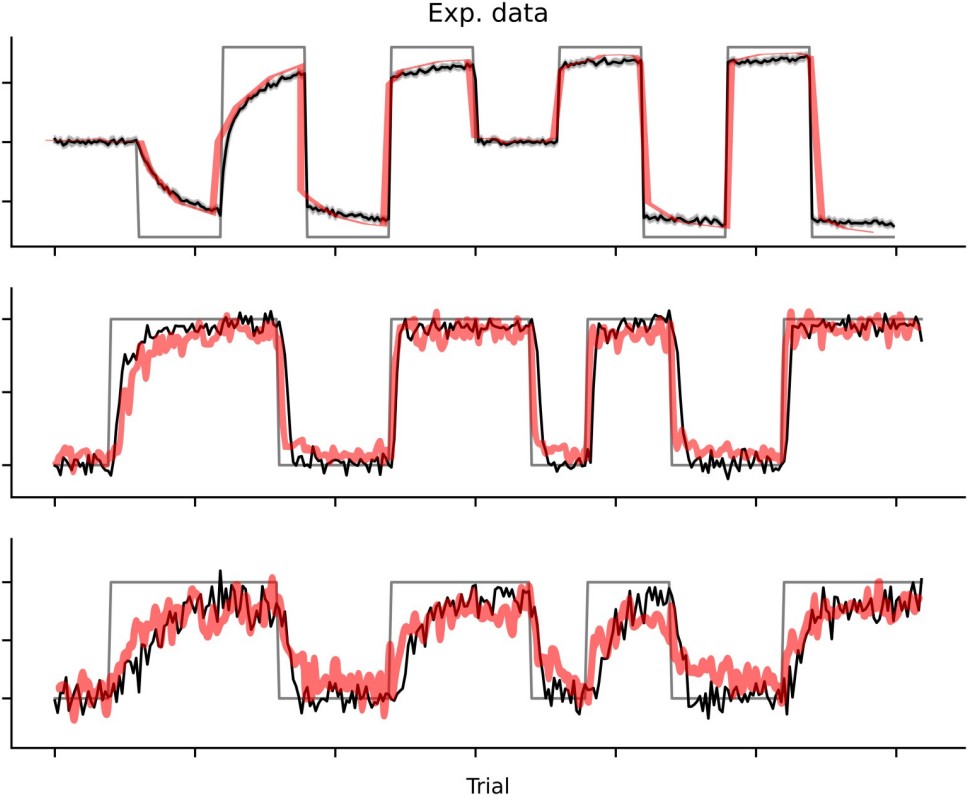

**Fig 4. Experimental results from [19] and figure 4A from [12] and simulations with the sCOIN model.** Only the measured adaptation is shown, where each panel represents one experiment. The simulations are shown in black and the experimental data in red. These results are the same as those from figure Fig 3, with the last two columns plotted together in one panel. (Top) Results from [19]. (Middle, bottom) Results from [12].

Schweighofer [12] analyzed savings during their second experiment and found that savings do exist, although greatly diminished compared to the first experiment. This is in favor of the dual-context model, as we present it here. In [12], a slow decay during error-clamp trials in their experiments was shown, which the authors considered evidence for the single-context account of the second experiment. However, as we show below, this can also be explained in a dual-context model as an effect of slow context inference.

**Uncertainty over contexts affects action selection.** As with learning, we show in this section that action selection is affected by context inference. If the identity of the current context is known, the forward model for this context is used to select the current action. However, if uncertainty over the context exists, the selected action is influenced by all the possible contexts, with a weight directly related to how likely each one of those contexts is (see Eq 6).

Experimental evidence supporting this view can be found in experiments with context switching. For example, Davidson et al. [1] reported a curl-force experiment in which participants had to switch between $A$ and $3A$ in one group, with a block sequence $A - 3A - A - 3A$, and from $A$ to $-A$ in another group, with a block sequence $A - (-A) - A - (-A)$. After $A$ and $3A$ (or $-A$ in the other group) had been learned in the first two blocks, the authors found that the switch from $3A$ to $A$ was faster than that from $-A$ to $A$. The authors interpreted this as evidence that switching between adaptations happens more quickly if it is in the same direction as the current adaptation (e.g. both counter-clockwise), and more slowly if they are in the opposite direction (e.g. clockwise to counter-clockwise).

Under the sCOIN model, the asymmetry is caused by the existence of the baseline context, which has a non-zero probability $p(\zeta_O|s_t \ldots)$, as can be seen in Fig 5A. When a new block of trials starts (e.g. in the transition from 3A to A), a switch is inferred by the model (given feedback after the first trial) and $\zeta_O$ becomes more likely (given that $\zeta_{3A}$ has been ruled out). Therefore, in these first trials, action selection has a component guided by the baseline model, in which no extra compensatory force is applied, effectively "pulling" adaptation towards zero (no compensatory force). In the first group, this initial pull towards zero accelerates the transition towards A because $3A > A > 0$, but in the second group, it slows down the switch because $A > 0 > -A$ and the behavior lingers around 0 until $p(\zeta_0|\ldots)$ drops back to zero.

To confirm this explanation, we simulated variants of the experiment in which the sCOIN model predicts that the difference between groups diminishes or disappears. First, in Fig 5B, we simulated an experiment in which the contexts have more extreme adaptations, making them more different from baseline than in the experiments in [1]. To do this, one group adapts in a $O - A - (-2A)$ paradigm, while the other group adapts in a $O - A - 4A$ paradigm. As in the original experiment, the second contexts ($-2A$ for one group, $4A$ for the other) are equally spaced from the first context. However, given the larger distance from baseline, the baseline context has the same probability for both groups after the switch back to A. This change makes

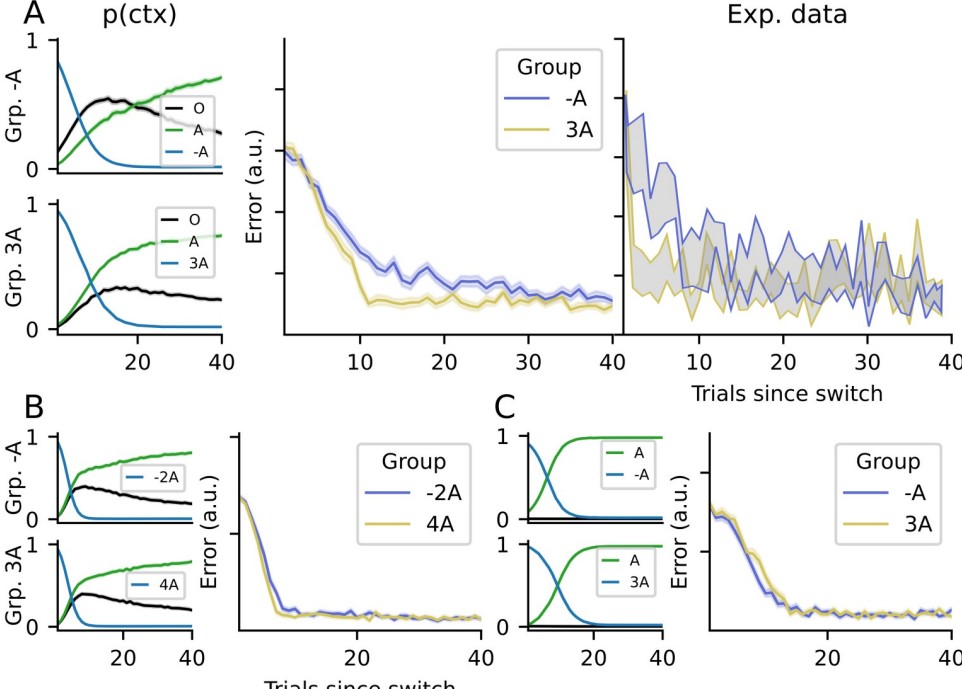

**Fig 5. Motor error when switching back to a previously-learned adaptation.** (A) Experimental results from [1] and simulations with the sCOIN model are shown. In the first column, each panel represents context inference for one group of participants (top: group $-A$; bottom: group $3A$), with each line representing the posterior probability of a context (black for the baseline O). The second column represents the error made by our simulated agent after returning to the previously-learned context, with blue and yellow representing groups -A and 3A, respectively. The last panel represents the same data from eight participants per group (mean ± SEM), from the experiment in [1]. All panels share the x-axis, representing the number of trials elapsed since the switch to the new context. All simulations were executed 32 times per group, to obtain a reliable mean; shaded areas represent the SEM across all simulations. (B) Simulated results for an experiment similar to [1], but changing the contexts seen by the two groups from -A to -2A for the first group, and from 3A to 4A for the second group. The panels follow the same structure as (A), without the last panel for experimental results. (C) Simulations for an experiment in which the baseline context has been removed altogether.

both simulated groups infer the correct context almost equally quickly, making the difference between their errors much smaller compared to the original experiment. Furthermore, we simulated an experiment with an identical structure to that in [1], but eliminated the baseline context from the agent. The results can be seen in Fig 5C, where the switches between contexts are made identically by the two groups.

**Action selection in error-clamp blocks.** During error-clamp blocks at the end of block sequences, participants' behavior can be divided in two phases: (1) Participants' behavior is consistent with a previously-encountered context (called spontaneous recovery in $O - A - B - E$ experiments, where behavior is consistent with $A$); this phase, when present, is seen during the early trials of the E block. (2) A slow return to baseline, which can last as long as hundreds of trials [25]. However, the direction of adaptation during the first phase, its duration, the delay before it is observed, the speed of the return to baseline and the final asymptote of the response vary greatly depending on the experiment [8, 16, 25, 26].

In this section, we show how context inference can explain these different parameters of behavior by changing the way contextual cues mislead participants' context inference, which in turn influences action selection.

This can be seen for example in [16], where the authors studied in detail human behavior during an error-clamp block in a shooting movement paradigm with a mechanical arm. The authors found that during an E block at the end of each experiment, there was a lag of a few trials (depending on participant) before their motor behavior changed from that of the previous block. After that, the exerted force slowly dropped towards zero throughout tens of trials, but never reaching values around zero. Participants were divided into four groups, each of which going through a different block sequence: (1.1) $A - E$, (1.2) $O - A - E$, (1.3) $(-A/2) - A - E$, and (1.4) $(-A) - E$. No pauses were made during the experiment nor were there any contextual cues, so transitions between blocks were not signaled to participants. However, because context inference integrates information from different sources, many experiments in which no intentional, overt contextual cues are available indeed contain contextual information that the participant can use to infer the context. For example, proprioceptive signals provide contextual information [7, 27]. The sudden appearance of motor errors can itself be a cue for contextual change [13] and even a pause between two trials could suggest a change in context [4].

In Fig 6A, we show data simulated with the sCOIN model, following the parameters of the experiment in [16], and in Fig 6B we show the experimental plots adapted from [16]. As in the experiments, we simulated error-clamp trials by forcing the observed error to zero, regardless of the action taken by the model. Following the experiments, 20% of trials in adaptation blocks were randomly set to error-clamp trials. The displayed adaptation is shown during the E trials for the three experimental groups in the experiment. It can be seen that group 1.3 (i.e. the participants who had learned in the $-A/2$ context in addition to $A$) more quickly recognized a change in context and lowered the force applied on the mechanical handle, as can be seen in the experimental data.

In Fig 6C, the inference over context is shown for each group separately. Context inference works reliably until the error-clamp trials start, which do not correspond to any of the known contexts. This causes the agent to infer the combination of some of the known contexts that best fits the observations. This is because the internal model for the inferred context predicts a spread of errors (see Eq 11), but observes all errors as zero because of the error clamp. As the error-clamp trials continue, action selection is made through a mix of both contexts, and the continued observation of zero error starts to favor the baseline context because it has a higher probability density of observing an error of exactly zero (since it has been practiced more by the participant/agent and the standard deviation is smaller for baseline than for adaptation contexts during the experiment). The behavior of the agent in the error-clamp trials depends

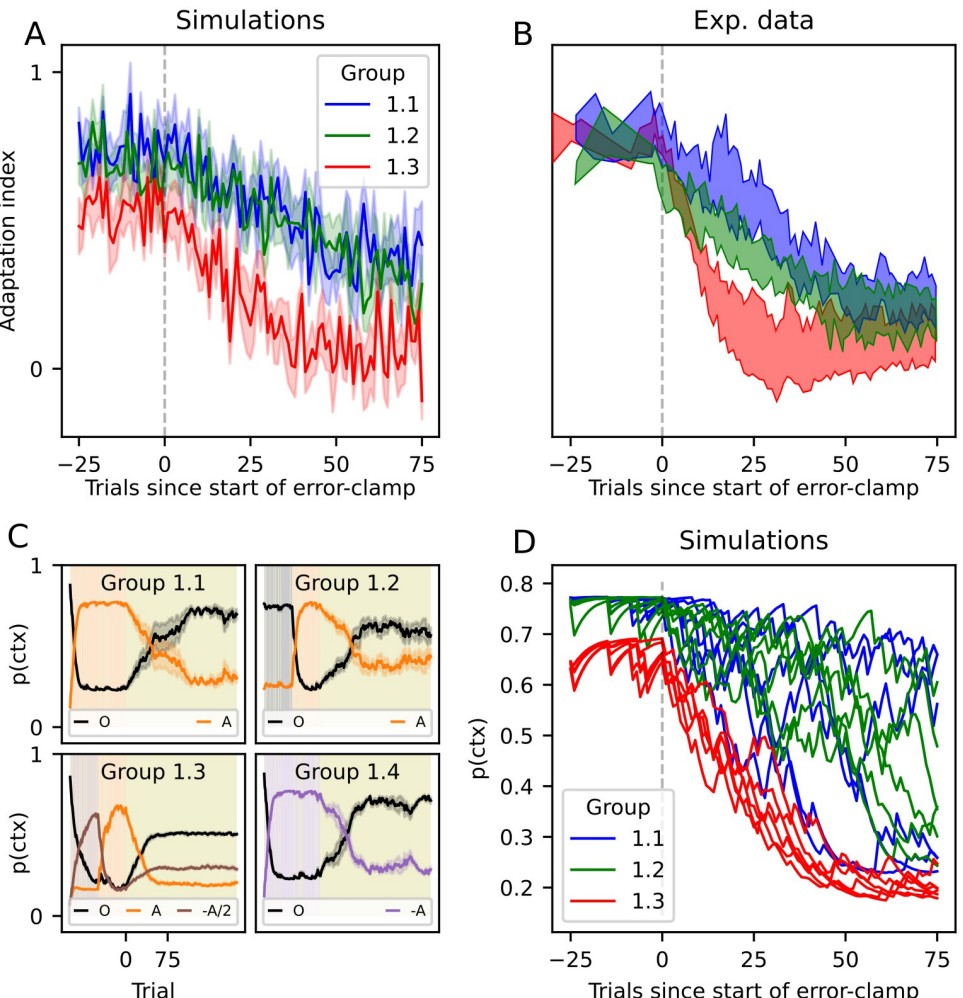

**Fig 6. Adaptation during error clamp trials.** (A) Simulated adaptation during the error-clamp trials for the three groups of participants in [16], using the same colors. Following [16], group 1.4 is not shown in A and B, as their behavior is identical to group 1.1. The data plotted are mean ± SEM across all participants (6 per group). The vertical dashed line is the start of the error-clamp trials. (B) Corresponding experimental data adapted from figure 2C in [16]. (C) Simulations: Inference over the current context, where contexts are color coded: black for baseline, orange for the counter-clockwise force, purple for the clockwise force and brown for counter-clockwise force with half strength. The lines represent the posterior probability of each context in every trial, while the background color represents the true context. Note that adaptation blocks include error-clamp trials, which are visible in the background. An olive-colored background represents error-clamp trials. As in (A), solid lines represent the average across all runs and shaded areas represent the standard deviation. (D) Simulations: Visualization of the lag before a change in context is detected by the agent during the *E* trials. Each line represents one run (10 runs per group).

on the contexts it previously learned: groups 1.1 and 1.4 display the same behavior, where the previous context (*A* and −*A*, respectively) slowly dwindles. These agents will slowly lower the force applied. In contrast, group 1.3 has learned the additional −*A*/2 context, which has a non-zero posterior probability during *E* trials, pushing the agent's adaptation force more quickly towards zero. Group 1.2 behaves similarly to 1.1, with the exception that the baseline context, which was recently seen, plays a bigger role during *E* trials, making the agent reduce its force during *E* trials slightly more quickly than groups 1.1 and 1.4.

In [16], the authors also found participant-specific delays before the decay to baseline started after the error-clamp phase began. As can be seen in Fig 6D, each simulated participant

follows a different path of decay, with variations caused directly by perceptual noise. In our simulations, however, it is clear that no such systematic lag can be directly observed, which is most noticeable when looking at context inference (Fig 6C), which begins the switch as soon as the error-clamp trials begin. This can be further observed in Fig 6D, where we plot each simulated participant (one run of the simulations, color coded as before); because observation noise was chosen randomly for each run, some runs appear to contain a large delay before the decay begins. This falls in line with the experiments in [25], who found that the lag observed in [16] disappeared when controlling for correlations in perceptual noise, as well as by using a balanced experimental design and unbiased analysis.

## Discussion

We showed that context inference as an active, continuous process, can explain many behavioral motor adaptation phenomena observed experimentally. In particular, we showed that the effects of the presence and reliability of contextual cues, as well as observation noise, can cause behavior that can be observed during context switching, as well as during times in which precise context inference is hindered, as is the case during error-clamp trials in many experiments.

To do this, we selected representative experimental studies that show the well-established effects of savings, spontaneous recovery and the effects of sensory cues. Using a simplified version of the COIN model introduced by Heald et al. [17], we showed how each of these effects can be explained by the dynamics of context inference, which integrates all the available information (e.g. sensory cues, workspace location, reward and endpoint feedback), in some cases throughout many trials.

With this, we expanded on previous works that introduced the idea that context inference is a process that informs and is informed by motor adaptation by showing that it explains behavioral phenomena that had previously required different specific, ad-hoc mechanisms outside of contextual motor adaptation.

### Further experimental evidence

In many cases, the context is not directly observable and context inference takes the form of an evidence-accumulating process that can take any amount of time to be certain of the context. It is in these cases where the effects of context inference are most noticeable. While many experiments exist that give probabilistic contextual information [28–30], evidence accumulation is not limited to these explicitly stochastic cases. Indeed, as we noted in the Results section, many experiments inadvertently include partial contextual information used by participants.

The most direct secondary contextual information comes in the form of reward and endpoint feedback. For example, participants may be told whether they obtained the desired reward at the end of a trial and are shown the end point of their movement. When participants observe an unexpectedly large error, they can infer that the inferred context might be incorrect. This is the case of the experiments in [12] shown in Fig 3B and 3C: if the adaptation is high, changes in context produce errors much larger than those of motor variability, and a context switch is easily and immediately identified; if adaptation is low, the errors produced by context switching are closer in magnitude to motor variability and evidence accumulation is necessary.

The same rationale explains the results in [13], as was shown in [17]: motor learning, which in the COIN model is modulated by context inference, is minimal for errors close to 2 and -2 (see their figure 2E). This is because an error of 2 or -2 signals that the participant incorrectly identified the context (as adaptation has a magnitude of 1). Additionally, as was shown in [17],

context inference explains the modulation of learning rate by the volatility of the environment observed in [13].

A subtler source of information can be found in long pauses between blocks of adaptation trials, after which an unprompted partial return to baseline has been observed [4]. This can be explained by context inference, as a long pause could prompt participants to infer that a switch had occurred, prompting participants to rely on their belief of the underlying probability of observing any of the known contexts, which is dominated by the previously observed context *A*, but now includes a component of the baseline *O*, as it is the most common one in everyday life.

Error-clamp (*E*) trials present another insight. If error is kept at zero, one could assume that participants would continue doing what they were doing before, as there is no reason (no observed error) to infer a change in context. However, this is almost never the case [4, 8, 9, 16, 31, 32]. Instead, a decay is observed, i.e. participants slowly reduce their adaptation, often first displaying spontaneous recovery [8]. Context inference provides a principled account of this behavior: the natural variability in participants' behavior lead them to expect errors, which clashes with the observed zero error. This prompts participants to re-evaluate their inferred context, which can partially activate a previously-observed context, as we showed in Fig 6. In [32], the authors found similar results, demonstrating that the duration of the previously-observed adaptation block also affects behavior in the *E* block. Additionally, in [33], the authors showed that introducing long periods before the E block begins lowers the initial force that participants exerted on the mechanical arm during the E block; longer periods of time make context inference revert to the prior expectation that a new baseline block begins, because participants are free to move their arm about during the pause.

In our account, if all information indicating a change in context is removed from the experiment, participants would continue to behave as they were in the previous block. Evidence for this can be seen in experiments 2 and 3 in [16], where participants were shown random errors during *E* trials, with a variance matching that of previously observed motor commands. The authors showed that by matching the errors expected by participants, they eliminated the slow tapering-off observed in most *E* blocks.

## Model predictions

The basic principle behind the results we presented is that the sCOIN model describes a process that develops over time and that carries with it uncertainty. This uncertainty affects learning and behavior during motor adaptation, effecting phenomena that are directly observable during behavioral experiments. In the following, we discuss several testable predictions that are direct consequences of the model.

For the model predictions discussed below, it is important to keep in mind that different contextual cues are not equally effective at separating motor responses during learning and switching [6, 24, 34, 35]. Because of this, the model predictions hinge on selecting the adequate type of contextual information that maximally helps the participants select the appropriate motor response.

**Error-clamp as a known context.** The inclusion of reliable sensory contextual cues (e.g. lights whose color uniquely identify a context) makes switching immediate, as in the experiments by [19]. We expect that the same effect would be observed in error-clamp trials. If the *E* block is learned by participants during training, it might still be difficult for them to infer that an *E* block has started, which would create delays similar to those in Fig 6. However, the model predicts that if a visual cue is introduced that uniquely and reliably identifies the *E* block, the posterior probability over contexts $p(\zeta_t = \text{EC})$ would be one, and participants would

immediately switch to their baseline behavior, no longer displaying an adapted response, lag, nor the slow return to baseline. This immediate switch in the presence of contextual cues would persist even if endpoint feedback is manipulated as was done in [16], as the inference is dominated by the cue component $p(\zeta_t|q_t)$.

**Interference effects during context switching.**   As discussed in the Results section, the effect observed in [1] is explained by the model as an effect of slow context inference drawn out over trials, instead of being a direct interference at the level of learning. As shown in Fig 5B and 5C, the context inference account predicts that this effect would disappear if all contexts were significantly different from baseline, such that the baseline context never explains the observations. Removing the baseline context from a participant's context inference might be experimentally unfeasible, but other possibilities include making all adaptations bigger (e.g. bigger angles, stronger forces), and including contextual cues that rule out the baseline context. In the opposite direction, the model predicts that if all adaptations are smaller (i.e. closer to baseline), the differences between the two groups would increase, although such differences might become impossible to detect in the behavioural data due to different sources of behavioural variations and noise.

**Multi-source integration.**   The model also predicts an effect reminiscent of multisensory integration [36]: in order to integrate contextual information from conflicting sources (e.g. probabilistic visual cues and noisy endpoint feedback), the weight placed on a source increases with its reliability. Such integration would manifest itself in experiments in which observations are noisy, as in the experiments in [11], in which the position of the finger was obscured and instead participants are shown a blurry cursor which was sometimes shifted from its true position. If the added observation noise gives evidence for a particular context (the true underlying context or another one) and a visual cue gave partial information for another context, the participants' behavior would be more consistent with the most reliable source of contextual information.

**Future work.**   In this work, we manually selected values for the model parameters to match the behavior seen in experiments. This approach served our purpose of making qualitative comparisons between the model's output and experimental results. These comparisons were sufficient to show that this model can adapt to multiple experimental setups which were not explicitly designed to test the model, and provide a unifying explanation to behavioral phenomena observed in those experiments.

One limitation of this approach is that it does not allow for a quantitative comparison between simulations and experimental data. We leave it to future works to address this limitation by explicitly fitting the model's parameters to participants' data in experiments specifically designed to test its predictions.

## Conclusions

The results we presented in this work indicate that several well-established behavioral phenomena observed across different motor adaptation experiments can be explained by the uncertainty in context inference and its effects on learning and action selection. Together with the results by [17], these results suggest new venues of investigation for future works in motor adaptation and context-dependent behavior.

## Methods

### The COIN and sCOIN models

In this work, we used a simplified version of the recently-introduced COIN model [17], adapted to the experiments that we covered in our simulations. In this section, we give a brief

introduction to the COIN model and, in the subsequent subsection, describe how we adapted the model to the experimental tasks. For a full description of the model, refer to [17].

**Generative model.** At each trial $t$, the agent infers both the context and the context-dependent adaptation (e.g. the parameters of the force field in mechanical-arm experiments). The context is represented by a latent, categorical variable $\zeta_t$, which is assumed to evolve over time according to:

$$p(\zeta_t | \zeta_{t-1}, \pi_{\zeta_{t-1}}) = \mathrm{Discrete}(\pi_{\zeta_{t-1}}) \tag{1}$$

where $\pi_{\zeta_{t-1}}$ is the transition probability vector from context $\zeta_{t-1}$ to all other contexts. The contextual cues (when present in an experiment) are assumed to be drawn depending on the context following:

$$p(q_t | \zeta_t, \Phi) = \mathrm{Discrete}(\Phi_{\zeta_t}) \tag{2}$$

where $\Phi_{\zeta_t}$ is the probability vector with which the contextual cue $q_t$ is shown to the agent in context $\zeta_t$. As pointed out by Heald et al. [17], both $\Phi$ and $\pi$ are in principle infinite, but a task-relevant finite set can be used instead.

The context-dependent adaptation is represented by the latent variable $x_{\zeta,t}$ and assumed to arise from an autoregressive process AR(1):

$$x_{\zeta,t} = a_\zeta x_{t-1} + b_\zeta + \omega_\zeta \tag{3}$$

where $a_\zeta$ and $b_\zeta$ are unknown, context-dependent parameters and $\omega$ is a Gaussian noise term of zero mean and unknown standard deviation $\sigma_{\zeta,x}$. This AR(1) process is assumed to have existed before the experiment begins and to have a stationary Gaussian distribution of unknown mean and variance:

$$p(x_{\zeta,t}) = \mathcal{N}(\mu_{\zeta,x}, \sigma_{\zeta,x}) \tag{4}$$

Note that $\mu_{\zeta,x}$ and $\sigma_{\zeta,x}$ are parametrized by the parameters of the AR(1) process, namely $\mu_{\zeta,x} = b_\zeta/(1 - a_\zeta)$ and $\sigma_{\zeta,x} = \sigma_q/(1 - a_\zeta^2)$, where $\sigma_q$ is a free parameter of the model which is not context dependent.

Observations take the form of state feedback (e.g. the position of the cursor on the screen in visuomotor rotation tasks), given by:

$$y_t = x_{\zeta_t,t} + v_t \tag{5}$$

where $v_t$ is a zero-mean Gaussian noise term with unknown standard deviation $\sigma_r$, which is a free parameter of the model.

Action selection (i.e. motor output $u_t$) is done via the weighted mean of $x_{j,t}$:

$$u_t = \sum_j p(\zeta_{j,t} | q_t...) x_{j,t} \tag{6}$$

where $p(\zeta_{j,t} | q_t...)$ is the predictive probability.

**Simplified COIN model.** The free parameters of this model can be fitted to participants' data, as was done in [17]. In this work, we instead chose values for these parameters to show that the model is capable of explaining the experimental phenomena in the Results section. Additionally, by fixing these parameters the agent is able to perform exact Bayesian inference at each trial using conjugate priors, replacing the MCMC approach used in [17] due to the mathematical intractability of the full formulation. This, however, does not significantly

change the model and was done purely for computational efficiency. In this section, we describe how we fixed parameters and the procedure for Bayesian inference.

As explained above, context is assumed to be a discrete variable which evolves as a Markov process. The transition matrices $\pi$ were generated via a Dirichlet process, with parameters that can be inferred from participant data ($\alpha$ and $\kappa$ in [17]). For a fixed value of these parameters, the transition matrices also become fixed. In our simulations, we set the probability of self-transitioning (denoted $p_\zeta$) depending on the experiments (see below), to numbers that approximate the experimental setup of each study.

Contextual cues are assumed by the agent to be sampled from a distribution that depends on the current context. This is done through a set of cue probability vectors that are generated via a parametric distribution, whose parameters are fitted to participants' data. For experiments that do not include probabilistic or deceiving cues, contextual cues, when present, unequivocally reflect the current context, i.e. $p(q_t = i|\zeta_t = j) = \delta_{ij}$, where $\delta_{ij}$ is the Kronecker delta, equaling one when $i = j$, zero otherwise. For the simulations of Fig 2, where contextual cues are probabilistic, cue uncertainty is implemented as $p(\zeta_t = i|q_t = i) = 1 - \eta$, where $\eta$ is the cue uncertainty, and $p(\zeta_t = i|q_t = j) = \eta/(N_\zeta - 1)\forall i \neq j$, where $N_\zeta$ is the total number of contexts in the experiment.

Using the above, the probability of a context for the state feedback for a trial after the cue has been observed is given by:

$$p(\zeta_t|\zeta_{t-1}, q_t, y_{1:t}) \propto p(\zeta_t|q_t)p(\zeta_t|\zeta_{t-1})p(\zeta_t|y_{1:t}) \tag{7}$$

where $p(\zeta_t|\zeta_{t-1})$ is given by the context self-transition ($p_\zeta$ in Table 1 below) such that:

$$p(\zeta_t = i|\zeta_{t-1} = j) = \begin{cases} p_\zeta & \text{if } i = j \\ \frac{1-p_\zeta}{N_\zeta-1} & \text{otherwise} \end{cases} \tag{8}$$

and

$$p(\zeta_t = i|y_{1:t}, a_{t-1}) = N(y_t - \hat{y}_i, \sigma_{\text{pred}}) \tag{9}$$

where $\hat{y}_i$ is the prediction made by context $\zeta_i$ given action $a_{t-1}$, obtained with the internal

**Table 1. Model and simulation parameters.** The star notation (e.g. $x_j^*$) denotes the real value used in the simulation of the task, which may be different from that assumed by the agent.

| | Var | Description | Kim (2015) | Oh (2019) | | Davidson (2004) | | Vaswani (2013) | | |
|---|---|---|---|---|---|---|---|---|---|---|
| | | | | Exp. 1 | Exp. 2 | Grp. 3A | Grp. -A | Grp. 1 | Grp. 2 | Grp. 3 |
| Task pars. | | Contextual cues | Yes | | | | No | | | |
| | $x_j^*$ | Adaptation sizes | 0, 40, -40 | 0, 20 | 0, 10 | 0, 4, -4 | 0, 4, 12 | 1 | 0, 1 | -0.5, 1 |
| | $\sigma_a$ | Adaptation noise | 0.01 | 1 | | 0.5 | | 0.1 | | |
| | $\sigma_r^*$ | Obs. noise | 3 | 2.5 | | 0.1 | | 0.1 | | |
| Agent pars. | $p_\zeta$ | Context self-transition | 0.9 | 0.98 | | 0.98 | | 0.9 | | 0.8 |
| | $\mu_0$ | Hyper priors | 0, -1, 1 | 0, 0 | | 0, 4, -4 | 0, 4, 12 | 0, 1 | | 0, 1, -0.5 |
| | $v_0$ | | 1e4, 1, 1 | 1e4, 1 | | 1e4, 1, 1 | | 1e4, 1 | | 1e4, 1, 1 |
| | $\alpha_0$ | | 25e3, 0.25, 0.25 | 22e3, 2.2 | | 33e3, 4e2, 4e2 | | 5e4, 5 | | 15e4, 5, 5 |
| | $\beta_0$ | | 1e5, 2, 2 | 1e5, 20 | | 1e5, 23e2, 23e2 | | 1e5, 2 | | 1e5, 2, 2 |
| | $\sigma_u$ | Motor noise | 1 | 2 | | | | 0.17 | | |
| | $\sigma_r$ | Obs. noise | 3 | 2.5 | | 2.5 | | 0.1 | | |

models (with the currently-estimated parameters) and $\sigma_{\text{pred}}$ is the prediction noise, which is given by

$$\sigma_{\text{pred}} = \sqrt{\sigma_\gamma^2 + \sigma_u^2 + \sigma_x^2} \tag{10}$$

where $\sigma_\gamma$ is the observation noise, $\sigma_u$ is motor noise and $\sigma_x$ is the standard deviation of the estimate of the parameters of the internal model (see below). The values of the observation and motor noises for each of the simulations can be seen in Table 1.

The internal models make predictions on future errors. The error predicted by the internal model of context $\zeta_i$ takes the form $\hat{y}_i = \epsilon_t + x_{i,t} + \omega(a_t)$, where $\epsilon_t$ represents the error at time $t$, $x_{i,t}$ is the estimate of the adaptation in context $\zeta_i$ (e.g. the angle of the visuomotor rotation) and $\omega(a_t)$ is the expected outcome of action $a_t$, e.g. the rotated reach movement. If the estimated adaptation $x_{i,t}$ matches the real value, and at trial $t$, $p(\zeta_t = i) = 1$, then the outcome of action $a_t$ and $x_{i,t}$ cancel out, leaving only $\epsilon_t$, which is zero for the experiments we simulate in this work. $\epsilon_t$ would only be non-zero in experiments where the error could compound from one movement to the next, such as the out-and-back movements from [1], but note that we only model the outward reaches in that experiment.

Finally, for the hidden variables $x_{j,t}$ we chose a stationary Gaussian distribution with unknown mean $\mu_x$ and standard deviation $\sigma_x$, instead of the AR(1) $a$ and $b$ parameters used in [17]. As a consequence, the sCOIN model does not have intrinsic memory decay, instead relying on the dynamics of context inference to explain the slow decay of memories during error-clamp trials [16, 25, 31].

Using Bayesian inference, the model infers the values of $\mu_x$ and $\sigma_x$ using a Gaussian likelihood and NormalGamma priors, which allowed us to use exact inference. The likelihood of the data is given by the prediction error of the observations:

$$p(y_t|x_t, \dots) = \mathcal{N}(y_t - \hat{y}, \hat{\sigma}) \tag{11}$$

where $\hat{y} = \sum_i \hat{y}_i p(\zeta_{t-1} = i|\dots)$ is the predicted observation given the previous observation and the previous action, and $\hat{\sigma}$ is the expected standard deviation of the predicted observation, given by the updated parameters of the model (discussed below).

We set priors over $\mu_{\zeta,x}$ and $\sigma_{\zeta,x}$ that enable exact inference over the latent variables $x$ (in what follows, we dropped the $j$ dependency for clarity):

$$\mu_x, \sigma_x \sim \mathcal{NG}(\mu_0, \nu_0, \alpha_0, \beta_0) \tag{12}$$

with free parameters $\mu_0, \nu_0, \alpha_0$ and $\beta_0$, which we fixed for each experiment separately. Because $x$ is context-specific, so are these parameters. This formulation comes with four free parameters (i.e. the hyper-priors $\mu_{0,i}, \nu_{0,i}, \alpha_{0,i}, \beta_{0,i}$), in accordance with the original formulation (note that Heald et al. [17] fixed the mean of the priors for $b$ to zero). While the two formulations are not mathematically identical, the effects of the hyper-priors for both are the same; we discuss these effects in the next section.

Because the likelihood function $p(y_t|x_t, \ldots)$ is Gaussian, this choice of priors allows us to calculate the update equations as follows:

$$
\begin{aligned}
\mu_{\phi,i}^{(t)} &= \frac{v_{\phi,i}^{(t-1)}\mu_{\phi,i}^{(t-1)} + p(\zeta_i|q_t, \ldots)s_t}{v_{\phi,t}^{(t-1)} + p(\zeta_i|q_t, \ldots)} \\[2mm]
v_{\phi,t}^{(t)} &= v_{\phi,t}^{(t-1)} + p(\zeta_i|q_t, \ldots) \\[2mm]
\alpha_{\phi,t}^{(t)} &= \alpha_{\phi,i}^{(t-1)} + p(\zeta_i|q_t, \ldots)/2 \\[2mm]
\beta_{\phi,i}^{(t)} &= \beta_{\phi,i}^{(t-1)} + \frac{p(\zeta_i|q_t, \ldots)v_{\phi,i}^{(t-1)}}{v_{\phi,i}^{(t-1)} + p(\zeta_i|q_t, \ldots)} \frac{(s_t - \mu_{\phi,t}^{(t-1)})^2}{2}
\end{aligned}
\tag{13}
$$

where $s_t$ represents the observations, in the form of the error between the observed and expected outcomes of the motor command. Note that the effect of the evidence (i.e. observations) on the inference over the context-dependent hidden states is gated by the probability of each context $p(\zeta_i|q_t, \ldots)$, as in [17].

To include motor noise (independent from estimation uncertainty), as well as carry over the uncertainty over $x_{j,t}$, we sample motor commands from a Gaussian centered on $u_t$ (see Eq 6), with a standard deviation $\sigma_u$, which is a free parameter of the model.

**Model parameters.** Table 1 lists all the parameter values that we used during our simulations. The parameters are divided into two categories: (1) task parameters, which encode the way we simulated the experimental design; (2) agent parameters, which correspond to the free parameters listed in the previous section. The variable names for the model parameters are given in the "Var" column, corresponding to the variables in the previous section. The values are divided into experiments and, within experiments, into the different groups or conditions that we simulated.

We estimated the task parameters from the information provided in their respective publications; when direct information was not provided, we estimated it from the reported results; these estimations are not exact, but function as a proof of concept. Agent parameter values are held constant for the different conditions or groups for each experiment, except those parameters that are expected to vary across conditions.

Because the sCOIN model does not have a mechanism for the online creation of new contexts, relying instead of a fixed number of contexts, the number of existing contexts was set according to each experiment. For the experiments in [19], to aid in learning of the two adaptations, the $\mu_0$ hyperparameters were set to -1 and 1 (plus the baseline of zero), which lead to the model learning the -40 and 40 visuomotor rotation angles, respectively. As the experiments in [12] have only one adaptation, this was not necessary and the new context was initiated with $\mu_0 = 0$. For the rest of the simulated experiments the focus was not on learning, but on the switching between known contexts, therefore we started simulations with models that had already learned the adaptations, setting the learned values to the real values used in each experiment.

The hyperparameters $\alpha_0$ and $\beta_0$ were set first for the baseline context such that the expected standard deviation of observations $\beta/\alpha$ roughly matched the observation noise in the task, i.e. $\beta_0/\alpha_0 \sim \sigma_r + \sigma_u$, while keeping the values for $\beta_0$ and $\alpha_0$ very high, which, together with the high $v_0$ values, ensure that learning in this context is very slow. For the other contexts, the ratio $\beta_0/\alpha_0$ was set to be higher than the baseline, while keeping the individual values $\alpha_0$ and $\beta_0$ much lower, to speed up learning.

The exact values for $\alpha_0$ and $\beta_0$ were set for each experiment such that $\beta_0/\alpha_0 = 2 * (\sigma_r \sigma_a)$, where $\sigma_a$ is the standard deviation of the adaptation. The rationale behind this choice is that $\sigma_r$ and $\sigma_a$ determine the noise in the observations made by the model at each trial, and their sum is the value of $\beta/\alpha$ to which the learning process converges with enough trials. We multiplied it by 2 in order to help in learning, specifically to make the *a priori* standard deviation higher for the untrained contexts than for the baseline context.

Of important note is the difference between the true observation noise and the expected observation noise in the simulations for the experiments in [1]. The expected observation noise $\sigma_r$ was set to a higher value to reflect the fact that feedback in curl-force mechanical arm experiments, while devoid of any added noise, is more difficult for people to use to inform adaptation than in other types of experiments due to the nonlinear nature of the force. This fact is reflected in the high number of trials necessary for full adaptation in these experiments as compared to, for example, visuomotor rotation experiments.

For the simulations in Fig 2, the parameters were set as in the experiments in [1], with two exceptions: (1) the cue uncertainty, which is set to the values of 0 and 0.33, for the low and high values, respectively; and (2) the agent's observation noise $\sigma_r$, with values of 0.5 and 2.

**Interpreting the hyper-parameters.** $\mu$ determines the initial estimate of the adaptation, in the same units as the necessary adaptation. $\nu$ encodes how stable this hyper-prior is: higher values (e.g. 10,000) all but guarantee that the hyper-prior $\mu$ will not change its value after observations; In principle, enough evidence should still modify it, but that would not happen during an experiment. Smaller values (i.e. $\sim 1$) make $\mu$ follow evidence more freely. Note that as more observations are accumulated, $\nu$ becomes bigger and bigger, stabilizing the value of $\mu$.

The hyper-parameters $\alpha$ and $\beta$ have a more complex effect. Note that the mean of a gamma distribution is $\beta/\alpha$; this mean is being used as the standard deviation of a Gaussian by the rest of the agent, which makes it an important measure of uncertainty. While setting the default hyper-parameters, the values used are, e.g., $\alpha = 0.5/\sigma_0$ and $\beta = 0.5$, where $\sigma_0$ is the *a priori* estimate of the standard deviation of the force exerted by the environment, which controls the initial learning rate. This makes the initial standard deviation equal $\sigma_0$, which makes it consistent with the fixed-force model. The 0.5 values ensure that uncertainty is large at the beginning and is greatly reduced during the experiment, but never to a point where it is so small that it makes trial-to-trial variation in the environment surprising. Changing this 0.5 would make the standard deviation change more quickly, making the model more or less precise in its predictions, independently of the volatility of the mean of the adaptation (via $\mu$).

The baseline model defaults to different values that make it a lot more stable, i.e. adaptation of the baseline context would require many more trials. The hyper-standard deviation of the mean is set to 10,000, which makes the mean entirely stable during the duration of the experiment. The values of $\alpha$ and $\beta$ are fixed regardless of $\sigma_0$ such that the standard deviation is 0.001 (compared that to the size of the adaptations in mechanical arm experiments, around 0.0125), and the hyper-parameters of the standard deviation are stable during the experiment.

## Author Contributions

**Conceptualization:** Dario Cuevas Rivera, Stefan Kiebel.

**Formal analysis:** Dario Cuevas Rivera.

**Investigation:** Dario Cuevas Rivera.

**Software:** Dario Cuevas Rivera.

**Supervision:** Stefan Kiebel.

**Visualization:** Dario Cuevas Rivera.

**Writing – original draft:** Dario Cuevas Rivera, Stefan Kiebel.

**Writing – review & editing:** Dario Cuevas Rivera, Stefan Kiebel.

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
