## [Decision Letter · Decision Letter 0]

8 Dec 2022

PONE-D-22-24673The effects of probabilistic context inference on motor adaptationPLOS ONE

Dear Dr. Cuevas Rivera,

Thank you for submitting your manuscript to PLOS ONE. After careful consideration, we feel that it has merit but does not fully meet PLOS ONE’s publication criteria as it currently stands. Therefore, we invite you to submit a revised version of the manuscript that addresses the points raised during the review process.

Thank you for submitting your manuscript to PLOS ONE. Your manuscript has been reviewed by two referees and their comments are appended below. As you will see, both reviewers note the merit of your work, but have expressed some concerns regarding accuracy, clarity analysis that preclude publication in PLOS ONE in the current form. Therefore, we invite you to submit a revised version of the manuscript that addresses the points raised during the review process.

In particular, both reviewers have noted inaccuracies and lack of clarity of the previously published experimental results that you wish to explain with the sCOIN model. Please provide more detail on the experiments (the figure suggested by Reviewer 1 is an excellent idea), and make sure that all terms are used and explained correctly (e.g., savings and error clamp), as Reviewer 2 notes.

The reviewers also felt that methods of the simulation experiments are lacking, and that quantitative analysis of the similarity of the simulated data and real subject data is required.

As both reviewers suggest, a discussion of the various model components in respect to their effects on the observed phenomena is missing.

Please also improve usability of the provided code, in accordance to the suggestions made by reviewer 1.

Finally, please take extra care to use correct terminology

We look forward to receiving your revised manuscript.

Kind regards,

Genela Morris, PhD

Academic Editor

PLOS ONE

Journal Requirements:

Funded by the German Research Foundation (DFG, Deutsche Forschungsgemeinschaft)

as part of Germany’s Excellence Strategy – EXC 2050/1 – Project ID 390696704 – Cluster

of Excellence “Centre for Tactile Internet with Human-in-the-Loop” (CeTI) of Technische

Universität Dresden.

However, funding information should not appear in the Acknowledgments section or other areas of your manuscript. We will only publish funding information present in the Funding Statement section of the online submission form. 

Funded by the German Research Foundation (DFG, Deutsche Forschungsgemeinschaft)

as part of Germany’s Excellence Strategy – EXC 2050/1 – Project ID 390696704 – Cluster

of Excellence “Centre for Tactile Internet with Human-in-the-Loop” (CeTI) of Technische

Universität Dresden.

Reviewers' comments:

Reviewer's Responses to Questions

**Comments to the Author**

1. Is the manuscript technically sound, and do the data support the conclusions?

Reviewer #1: Yes

Reviewer #2: Partly

2. Has the statistical analysis been performed appropriately and rigorously? 

Reviewer #1: N/A

Reviewer #2: No

3. Have the authors made all data underlying the findings in their manuscript fully available?

Reviewer #1: Yes

Reviewer #2: Yes

4. Is the manuscript presented in an intelligible fashion and written in standard English?

Reviewer #1: Yes

Reviewer #2: Yes

5. Review Comments to the Author

Reviewer #1: Summary:

This work proposes a simplified version of the recently introduced "context inference" (COIN) model (Heald et al. 2021) for modeling context dependent motor adaptation. In contrast to the original COIN models, the simplified model (termed “sCOIN”) retained the mechanisms for context inference, motor adaptation, and action selection while eliminating the components that support learning of new contexts and estimating the dynamics of environment transition between contexts. The authors demonstrate that sCOIN is capable of displaying three contextual motor learning phenomena, unexplored by Heald et al. (2021) but previously investigated in experimental studies on motor adaptation. These phenomena are linked to the effect of contextual information (cues and feedback) on context inference which in turn affects switching behavior. The model was utilized to simulate the learner adaptation behavior rather than utilizing it to fit subject data. The study shows that in simulations of the original setups, the "simplified COIN" (sCOIN) model exhibits motor adaptation sequences qualitatively similar to real subjects shown in the original papers.

Strengths:

(1) Although I don't have much experience with motor learning modeling, the suggested simplification of the COIN model appears intriguing and the paper demonstrates its potential to explain interesting phenomena.

(2) The model's behavior in simulations is very similar to experimental data.

Weaknesses:

(1) The paper is well written but some parts are difficult to understand. The reader must refer to the original studies to understand the experimental paradigms. I would recommend adding illustrative figures of the paradigms and emphasizing the nature of cues and feedback.

(2) The link between the paradigm aspects (both environment and learner) and the model variables is not always clear. In addition, how exactly the specific components of the model give rise to the observed phenomenon is not always explained (see detailed comments below).

(3) The similarity between the simulated and real data is compared only qualitatively. I wonder whether these similarities could also be captured quantitatively.

(4) It is not clear to me (Despite the reasons given in the paragraph starting in line 523) why the authors did not try to fit the sCoin model parameters to real subjects data instead of arbitrarily/empirically selecting them for the simulation.

Detailed comments:

Line 66: ungrammatical "In this work show that"

Line 125: typo “alread-learned”.

Line 136: provide more information on the "generic experiment". It would be easier to understand how contextual cues and sensory feedback are provided and noised. (Instead of just referring the reader to Davidson and Wolpert (2004))

Figure 1: In panel A mention that this is the Schematic representation of the sCOIN model (rather than COIN Model).

Figure 1: In the left figure of panel B, if the context changed at trial 20, It is not clear how the inferred state (force) does not change even in the high certainty and low noise setup? It may be due to the fact that I do not understand the experiment setup. Please clarify.

Line 218: which one is the center panel in figure 2A? There are 4…

Line 245: I am missing here the link between the observed data (both real and simulated) and the model. Which component/s of the model are responsible for the main observed phenomena? It would help the reader better understand the model workings and appreciate its ability to exhibit those behaviors (similar to how it was done later on in the paragraph starting from line 274.)

Line 250: typo “ever” instead of “every”.

Line 289: typo "O" instead of "0".

Figure 2: How many agents/runs were included in the simulation and in the original experiment? How model/simulation parameters were set? Mention that it is detailed later on in the paper.

Figure 2: What does the gray region around the black line in the third and fourth panels represent? Why there is no black line in the third column of 2A as opposed to the other experiments (2B and 2C).

Figure 2: No blue lines in the figure. Do you mean orange?

Figure 3: Use the model name consistently throughout the paper, either sCOIN or SCOIN.

Figure 3A: In the Davidson and Wolpert (2004) paper, as far as I understand the post switch experiment length is 40 trials not 60. Can you elaborate on the difference?

Figure 3 caption: in the 4’th line, Group -A instead of group A.

In Figure 4C group 1.2, it is not clear why the posterior for the baseline context starts from 0.5 while in the other groups it starts with 1.

Line 338: It would help if you explicitly explain what parts of the sCOIN model cause the increase in the probability of the baseline context in EC trials.

Line 452: Base this prediction on the mechanisms of the model, beyond the explanation that it is allowed by the process of context inference.

Line 542: use consistent notation: Either \\delta or d; either c or \\zeta.

Code: No package requirement file is provided (instead only a dump of an environment suitable for Linux) making it difficult to run the project on other OS.

Code: It is not clear how to run the code in the repository. Please specify the entry point/s and add other needed information about how to run the project to reproduce the simulation results presented in the paper.

Reviewer #2: Reviewer summary: Recently, computational models and Bayesian for motor adaptation have been introduced to demonstrate the effects of context inference on learning rates in various experiments. Cuevas Rivera, Darío and Kiebel built a simplified version of the recently introduced COIN model to demonstrate that the effects of context inference on motor adaptation and control extend even further than previously demonstrated. They used this model to simulate previous motor adaptation experiments and demonstrated that context inference, and how it is affected by the presence and reliability of feedback, affect a variety of behavioral phenomena which had previously required multiple hypothesized mechanisms. The authors suggest that the reliability of direct contextual information, as well as noisy sensory feedback, which is common in many experiments, cause measurable changes in switching-task behavior and action selection that are directly related to probabilistic context inference.

I think the manuscript in its current state is a good starting point. The overall clarity of the writing needs improvement. However, I feel as though the authors lack a clear understanding of the behavioural phenomena they are attempting to model. Savings is not the absence of learning upon re-exposure to the same previously experienced perturbation, but faster relearning. Error clamps are not a zero force manipulation as in baseline trials. There is a matching of the force generated by participants and visually displayed to participants as zero error. Any previously learned perturbations will decay at this block of trials, rather than a learning of zero error.

for minor and major issues, please see attached pdf

6. PLOS authors have the option to publish the peer review history of their article (what does this mean?). If published, this will include your full peer review and any attached files.

Reviewer #1: No

Reviewer #2: No

---

## [Author Response · Author response to Decision Letter 0]

28 Feb 2023

We have responded to all comments, suggestions and questions made by the reviewers in the attached file. We have also addressed all of the editor's comments.

---

## [Decision Letter · Decision Letter 1]

7 May 2023

PONE-D-22-24673R1The effects of probabilistic context inference on motor adaptationPLOS ONE

Dear Dr. Cuevas Rivera,

Thank you for submitting your manuscript to PLOS ONE. Your revised manuscript has now been two reviewers, original Reviewer 1 and a new Reviewer 3. Unfortunately, Reviewer 2 was not available to assess your revision. As you will see, Reviewer 1 was satisfied with your revision. Reviewer 3's review was favorable, but they suggested a few corrections that we feel will help further improve the presentation (see in the attached annotated pdf). Therefore, we invite you to submit a revised version of the manuscript that addresses these points.

We look forward to receiving your revised manuscript.

Kind regards,

Genela Morris, PhD

Academic Editor

PLOS ONE

Journal Requirements:

Reviewers' comments:

Reviewer's Responses to Questions

**Comments to the Author**

1. If the authors have adequately addressed your comments raised in a previous round of review and you feel that this manuscript is now acceptable for publication, you may indicate that here to bypass the “Comments to the Author” section, enter your conflict of interest statement in the “Confidential to Editor” section, and submit your "Accept" recommendation.

Reviewer #1: All comments have been addressed

Reviewer #3: (No Response)

2. Is the manuscript technically sound, and do the data support the conclusions?

Reviewer #1: Yes

Reviewer #3: Yes

3. Has the statistical analysis been performed appropriately and rigorously? 

Reviewer #1: I Don't Know

Reviewer #3: Yes

4. Have the authors made all data underlying the findings in their manuscript fully available?

Reviewer #1: Yes

Reviewer #3: Yes

5. Is the manuscript presented in an intelligible fashion and written in standard English?

Reviewer #1: Yes

Reviewer #3: Yes

6. Review Comments to the Author

Reviewer #1: The authors have addressed my comments in a satisfactory manner. I have no further major comments on the paper.

I would like to commend the authors for their efforts in revising the manuscript and improving the clarity and rigor of their work.

Reviewer #3: In this manuscript, the authors build on a previously reported COIN model (Heald and Wolpert 2020) that describes how contextual information learns from and informs motor adaptation. The authors make a generalization of this model which not only advances the field but also broadens the scope of this model

7. PLOS authors have the option to publish the peer review history of their article (what does this mean?). If published, this will include your full peer review and any attached files.

Reviewer #1: **Yes: **George Kour

Reviewer #3: No

---

## [Author Response · Author response to Decision Letter 1]

11 May 2023

Responses to reviewers' comments are included as a pdf file

---

## [Editor Report · Decision Letter 2]

23 May 2023

The effects of probabilistic context inference on motor adaptation

PONE-D-22-24673R2

Dear Dr. Cuevas Rivera,

We’re pleased to inform you that your manuscript has been judged scientifically suitable for publication and will be formally accepted for publication once it meets all outstanding technical requirements.

Kind regards,

Genela Morris, PhD

Academic Editor

PLOS ONE

---

## [Editor Report · Acceptance letter]

21 Jun 2023

PONE-D-22-24673R2 

The effects of probabilistic context inference on motor adaptation 

Dear Dr. Cuevas Rivera:

I'm pleased to inform you that your manuscript has been deemed suitable for publication in PLOS ONE. Congratulations! Your manuscript is now with our production department. 

Kind regards, 

on behalf of

Dr. Genela Morris 

Academic Editor

PLOS ONE